# Integrated Safety and Health Promotion among Correctional Workers and People Incarcerated: A Scoping Review

**DOI:** 10.3390/ijerph20126104

**Published:** 2023-06-12

**Authors:** Olivia J. Hull, Olivia D. Breckler, Lisa A. Jaegers

**Affiliations:** 1Occupational Therapy Doctorate Program, Johnson & Wales University, Providence, RI 02903, USA; livjhull@gmail.com; 2Department of Occupational Therapy, Findlay University, Findlay, OH 45840, USA; brecklero@findlay.edu; 3Department of Occupational Science and Occupational Therapy, Saint Louis University, St. Louis, MO 63104, USA

**Keywords:** total worker health, incarceration, recidivism, antisocial behavior, prosocial behavior, corrections, work environment

## Abstract

Improving safety and health for correctional workers and people who are incarcerated are widespread yet separate initiatives. Correctional workers and people who are incarcerated experience similar challenges involved with poor workplaces and living conditions, including mental health crises, violence, stress, and chronic health issues, and the available resources lack integration with respect to safety and health promotion. This scoping review sought to contribute to an integrated approach for correctional system safety and health resources and identify studies of correctional resources that address health promotion among correctional workers and people who are incarcerated. Guided by PRISMA, a search of gray literature, also termed peer-reviewed literature, published between 2013–2023 (*n* = 2545) was completed, and 16 articles were identified. Resources primarily targeted individual and interpersonal levels. At every level of intervention, resources improved the environment for both workers and those incarcerated, with trends of less conflict, more positive behaviors, and improved relations, access to care, and feelings of safety. The corrections environment is impacted by changes from both workers and people who are incarcerated and should be examined using a holistic approach. Future health and safety resources should target the larger correctional environment by utilizing practices, policies, and procedures to improve safety and health for incarcerated people and workers.

## 1. Introduction

The latest data report shows that there will soon be 11.5 million people who are incarcerated worldwide [1]. The increasing numbers of people who are incarcerated, paired with high levels of stress, violence, and mental health issues within the correctional systems pose a significant global public health issue [2,3,4,5,6]. The United States has one of the highest total prison populations, with 1.7 million people [1], in addition to the 6.9 million people cycling through jails, and the 5.4 million people on probation or parole [7,8,9]. Correctional systems face challenges that span across staff and people who are incarcerated (e.g., those in jail or prison, or on probation or parole) including mental health crises, chronic health issues, and high rates of incarceration, paired with low staffing and high turnover [2,3,4,5]. Challenges in corrections including overcrowding, violence, and stress impact both those incarcerated and the workers. For correctional workers to ensure the well-being and safety of people who are incarcerated, they must address their own well-being and health as well [10]. Correctional workers report high rates of mental health disorders (~30%) and work-related stress (22–35%) [2,11]. Correctional officers also experience high rates of violent acts against them in the workplace [12,13] and poor workplace conditions, such as inadequate pay, mandatory overtime, and understaffing [14].

People who are incarcerated also face comparatively higher levels of mental and physical health effects from being confined, which can be compounded by increased use of solitary confinement and social isolation [15,16,17]. People who are incarcerated face additional challenges during reentry including housing, transportation, employment, and social relationships [18,19]. They also face a risk of rearrests—with 50% of those formerly incarcerated being rearrested after 3 years [20].

The purpose of corrections has evolved from a punitive approach and is shifting towards a rehabilitative approach with the goal of providing safer environments for staff and those incarcerated [21,22,23]. Nationally, there are calls for change by the Federal Bureau of Prisons (FBP) and the National Occupational Research Agenda (NORA) for providing safer environments in correctional facilities [24,25]. In an effort to consider the health of both workers and those detained in corrections facilities, integrated interventions that address both the workforce and people who are incarcerated is necessary.

Considering the health promotion needs across correctional systems is complex, and an integrated approach is needed to consider prevention targets, levels of influence, and integration of health-hazard protections. One strategy to address the workplace, internal and larger communities is the National Institute for Occupational Safety and Health (NIOSH) Total Worker Health^®^ (TWH). A TWH approach considers policies, programs and practices to improve workplace safety, health, and well-being [26]. The TWH strategy considers interventions at social ecology levels, including individual, interpersonal, community, organizational, and policy [27,28]. Furthermore, TWH applies a hierarchy of controls to prioritize interventions and emphasizes the use of organizational-level interventions [26]. The most-effective controls are those designed as either primarily preventive or to eliminate causal conditions within workplaces that place individuals at risk for illness and injury; the next most-effective control is replacing unhealthy conditions with those that promote health through policies and upper-level practices [26]. The least-effective controls are those designed as reactionary (secondary and tertiary interventions) and for the individual level [13,26]. However, when integrated across levels and types of controls, each type of intervention is needed due to the high prevalence of physical and mental illnesses among both workers and people who are incarcerated [13].

Reviews (e.g., systematic, scoping, and meta-analysis) have been completed on health promotion interventions solely for correctional workers [29,30,31,32] and solely for people who are incarcerated [33,34,35]; however, this study further expands upon investigation of health and promotion interventions for both workers and people who are incarcerated. A gap exists in the occupational safety and health and criminal justice literature addressing health promotion resources that consider both correctional workers and people who are incarcerated. The purpose of this scoping review study was twofold: (1) to contribute to the understanding of an integrated approach for correctional-system safety and health resources, and (2) to identify studies of correctional resources that have addressed both correctional workers and people who are incarcerated safety and health promotion.

## 2. Materials and Methods

This scoping review of existing literature was conducted following PRISMA—Preferred Reporting Items for Systematic Reviews and Meta-Analysis’ extension for scoping-review guidelines [36] and utilized the JBI *Manual for Evidence Synthesis* [37]. A protocol was created by the research team and can be accessed by contacting the first author.

### 2.1. Eligibility Criteria

Inclusion and exclusion criteria were developed through consultation with the research team and eligibility criteria were based on the inclusion/exclusion criteria listed in Table 1. Articles were required to: address both workers of correctional facilities and people who are incarcerated; describe intentional examination of the intervention’s impact on both those incarcerated and workers; and demonstrate distinct outcomes. To contribute to an integrated approach, the articles were excluded if they did not address outcomes for both workers and people who are incarcerated. The overall study had to be within the carceral system, and all geographical areas were considered. Additional criteria included: full-text studies in English or that could be translated into English; published between 2013 and 2023; peer-reviewed articles of any research study design; and thesis/dissertations, gray literature, or white papers. A wide range of article types were examined to obtain comprehensive knowledge on the topic. Reviews (e.g., systematic, scoping, and narrative) were examined to identify if any articles within the review met the inclusion criteria if not already identified in the initial search. This review examined all resources offered for incarcerated people or workers, and they were included in this study as the intervention source.

### 2.2. Search Strategy

Search terms were developed by considering a recent National Institute of Corrections scoping review of correctional worker health resources [29] and performing a preliminary search to determine commonly used terminology in criminal justice literature. Consultation with the research team and a research librarian was completed to finalize the search string in Appendix A.

### 2.3. Information Sources

The search for evidence started on 23 January 2023 and ended 27 January 2023, using databases including: Academic Search Complete (EBSCO), CINAHL Complete, PubMed, Criminal Justice Abstracts (EBSCO), APA PsycArticles, and ERIC. Searches for gray literature and white papers included: Google Scholar, the National Institute of Corrections’ library, Vera, the Prison Policy Initiative, the Correctional Leader Association, and the National Institute of Occupational Safety and Health Library. Due to the nature of searching gray literature with the maximum word limits, an adapted search string was created, which can be found in Appendix B. Furthermore, the reference lists of articles and articles used to identify keywords were examined for eligibility.

### 2.4. Selection of Sources of Evidence

Screening by title and abstract against the inclusion criteria was completed by two independent researchers. Prior to the screening, both reviewers screened the same 20 publications and compared their results to improve inter-rater reliability. Two independent reviewers completed full-text screening and compared their findings for discrepancies. A third researcher provided consensus review to resolve conflict. A data-charting tool designed by the first researcher was used to extract and capture the relevant information on key study characteristics (Appendix C). Similarities in the level of intervention were analyzed and grouped thematically by social ecology level. Descriptions of interventions and outcomes were extracted and examined by the first author.

## 3. Results

### 3.1. Selection of Sources of Evidence

As shown in the PRISMA flowchart (Figure 1), 2545 abstracts were screened on the basis of the inclusion criteria; 2446 were excluded. The remaining 99 articles were read in full by the first two authors, and 87 articles were excluded because either they did not have an intervention, they lacked outcomes for both those incarcerated and workers, they were not in the correct year-range, they considered different populations, or the study could not be retrieved. 12 articles remained for full data extraction. Four reviews were analyzed [38,39,40,41], from which 14 articles were identified that could meet the review and were extracted into full texts to be assessed for eligibility. Of those, 10 articles were excluded, and four articles were included in this review. Ultimately, a total of 16 articles met inclusion criteria and were summarized in this scoping review.

### 3.2. Characteristics of Sources of Evidence

Included articles (*n* = 16) are summarized in the data extraction in Table 2. Study designs represented were the quasi-experimental (*n* = 3) [42,43,44], qualitative (*n* = 3) [45,46,47], mixed-methods (*n* = 2) [48,49], pilot study (*n* = 2) [50,51], program evaluation (*n* = 1) [52], embedded case study (*n* = 1) [16], content analysis (*n* = 1) [53], quantitative analysis (*n* = 1) [54], phased intervention and quasi-randomized controlled trial (*n* = 1) [6], and pragmatic exploratory trial (*n* = 1) [55]. Most research was conducted in the USA (*n* = 9) [16,42,44,46,48,50,52,53,54] and the UK (*n* = 4) [6,45,51,55]. Two studies were from Canada [43,49], and one from Australia [47]. Types of study-settings included community supervision (*n* = 6) [42,43,44,50,52,55], prison (*n* = 7) [45,46,47,49,51,53], community supervision and prison (*n* = 1) [6], correctional center (*n* = 1) [48], and juvenile justice facility (*n* = 1) [54]. The intervention was introduced through programs (*n* = 7) [6,46,48,49,50,52,53], education/training (*n* = 4) [42,43,44,55], models (*n* = 3) [47,51,54], and policies (*n* = 2) [16,45].

### 3.3. Synthesis of Results

#### 3.3.1. Individual Workers

Three interventions were targeted at the individual level for workers; they measured outcomes for both workers and people who were incarcerated [43,44,55]. Results show that when officers were targeted at the individual level and were given an opportunity to improve skills or knowledge, there was a beneficial outcome for the people who were incarcerated as well. One used a novel intervention termed ‘psychologically informed practice’ (PIP) for staff working with people on probation who had personality disorders [55]. The staff who went through the training reported significantly higher levels of knowledge and understanding regarding personality disorders and increased feelings of accomplishment, as compared to the comparison group. The probationers had significantly lower rates of recalls (going back to prison if a probation rule is broken) and warnings, compared to a group that did not have the intervention. The other two articles focused on probation/parole training for officers [43,44]. The Strategic Training Initiative in Community Supervision (STICS) for probation officers was utilized to distinguish criminogenic needs (e.g., risks associated with recidivism) from non-criminogenic needs [43]. They also used cognitive and behavioral techniques to alter or deter the seeing of the benefits of completing crimes; the latter is termed a ‘pro-criminal attitude’ [56]. After this training, probation officers discussed with their clients significantly more criminogenic needs, such as pro-criminal attitudes, and had significant improvements in collaborative skills, session structuring skills, and cognitive/behavioral techniques. Probationers had significantly lower general-recidivism rates. Similarly, Latessa et al. (2013) [44] focused on a training termed EPICS (Effective Practices in Community supervision), intended for probation and parole officers. Officers who received the training were more likely to focus on criminogenic needs and recognize a link between thoughts and behaviors. Officers who used more of the training techniques also had a better-perceived relationship with their clients, as well as less arrests. The clients who were classified as moderate-risk perceived the relationship with their officers to be more caring, trusting, and fair. High-risk clients perceived the relationship to be tougher. The clients who perceived a trusting relationship with their officers were significantly less likely to be arrested for a new crime.

#### 3.3.2. Individual Incarcerated People

Improved prosocial behaviors and the acquisition of knowledge and skills were themes in these individual-level intervention articles targeted toward people who were incarcerated. Due to the incarcerated people’s demonstrating fewer anti-social behaviors, the workers were directly impacted as well. Four interventions were targeted at an individual level for people who were incarcerated, and measured outcomes for both workers and those incarcerated [46,48,51,53]. Two articles focused on prison dog training programs [46,53]. Doyon-Martin and Gonzalez (2022) [53] focused on how a prison-based dog training program impacted four different Midwestern U.S. prisons. Out of the staff who contributed responses regarding the impact of the program, 25% thought that the program contributed to less violence and improved interactions in the facility, and 34% believed that the program led to an overall positive attitude in the facility. Out of the incarcerated participants, 29% reported a rehabilitative impact, and 21% reported improved self-esteem, parenting skills, and more hope for the future. One-third of the respondents used the exact phrase that they felt as if they had a “sense of purpose” and 100% of those incarcerated were able to identify at least one personal benefit from the program. Likewise, Jones (2018) [46] reported less-aggressive communication between staff and those incarcerated as well as increased self-esteem, responsibility, and self-control, and an overall positive, calm environment for the people in prison after the implementation of a prison-based dog program. Both articles noted that the program “bridged a gap” between officers and those incarcerated and reported improved vocational skills for future employment [46,53].

Miller (2014) [48] utilized a reentry program to understand the collateral benefits of reentry initiatives and focused on addressing issues that people who are incarcerated will face post-release. Participants reported positive feedback on the program and specifically enjoyed having structure because it gave them purpose. Participants also reported solidarity between group members. The workers saw a significant reduction in altercations, such as the uses of force, fights, assaults, and disturbances. Some officers reported that the culture of the facility had improved since the program started. Camp et al. (2018) [51] examined the effectiveness of the Enhanced Support Model, which consisted of individualized psychosocial interventions based on cognitive behavioral therapy (CBT), motivational interviewing, emotional regulation, goal setting, and interpersonal effectiveness. Staff perceived that the intervention led to better coping, understanding and management of behaviors, and an increase in prosocial behaviors. Those incarcerated validated this by showing more positive behaviors and less aggression and noncompliance.

#### 3.3.3. Individual Incarcerated People and Workers

Two interventions were targeted at an individual level for both people who are incarcerated and workers [6,47]. Davies et al. (2021) [6] investigated the outcomes of a brief mindfulness training intervention on staff and people in prison or on probation. Both prison staff and prison residents’ knowledge and skills as to mindfulness significantly improved, and they reported less stress after the intervention. Neuhaus et al. (2018) [47] explored the effectiveness of Hepatitis C telemonitoring from the perspectives of clinical staff and people who were incarcerated. A total of 108 people who were incarcerated commenced treatment for Hepatitis C, with 75% completing the full treatment. Staff reported improved knowledge and confidence in treating Hepatitis C, as well as being able to provide greater access to care.

#### 3.3.4. Interpersonal

Interventions targeted at the interpersonal level focus on creating change in individuals through social influence and changing existing social relations [28]. Interpersonal interventions showed improvement in officer–client relationships and access to care, as well as fewer violations. These interventions have the potential to improve the overall social environment. Three interventions were targeted at the interpersonal level, therefore impacting both workers and people who are incarcerated [42,49,52]. Manchak et al. (2014) [42] utilized a special mental health training for probation officers. The specialty officers demonstrated higher compliance skills and were more effective in accessing psychiatric services. Probationers with specialty probation were less likely to have formal violations and reported a better relationship with their officers. Bardin et al. (2022) [52] evaluated a gender-matched probation program (FOCUS),e where women probation officers implemented women-only caseloads in order to address mental health and connect women to community services and comprehensive medical care. The officers reported that the training helped them acknowledge their clients’ successes and the decreased caseload allowed them to develop a relationship with their clients and restore trust in the system. Additionally, clients reported they had good relationships with their officers and that their probation officer helped them access mental health care, food, and employment, and improved their relationships with their kids. Lastly, 50.4% of clients received family welfare services, and 69% received health insurance.

Within a correctional center, Weinrath et al. (2021) [49] examined differences in behaviors and alternative resolutions used in a therapeutic community, as compared to other units. The therapeutic community provided a structure for those in prison through work, routines, leadership roles, and responsibilities. The officers were trained with an emphasis on interpersonal relations and supporting the goals of the community. The therapeutic communities had less misconduct compared to other units and the officers had to complete fewer forced moves. Officers were more likely to use alternative resolutions and reported more interactions that supported a stronger relationship. Residents in the therapeutic communities reported that the officers were more likely to listen and talk more, which led them to want to have increased discussions with the officers.

#### 3.3.5. Organizational

At the organizational level, studies addressed people who were incarcerated and workers with interventions that led to increased feelings of safety and overall health [50,54]. They focused on creating an organizational change to encourage or support positive behavioral changes among employees or people who were incarcerated [28]. Lichtenstein and Barber (2016) [50] conducted an HIV education program and hosted on-site HIV services for probationers and parolees. After the programming, 100% of the officers reported having more knowledge of HIV, 88% of the officers reported being less fearful about becoming HIV-infected, and 63% reported being less concerned about supervising clients who had HIV. Additionally, among those newly incarcerated, all (*n* = 86) received HIV education, 37% volunteered for pretest counseling, and 31% volunteered to be tested and receive their results. Of the 249 currently incarcerated, 12% volunteered for pre-test counseling, and less than 1% volunteered to be tested and receive their results. Elwyn et al. (2015) [54] examined the implementation of the Sanctuary model, an evidence-based, trauma-informed care intervention, and its impact on the physical/psychological safety of staff and youth at a female juvenile facility. Incarcerated youths and staff both reported feeling safer after the implementation of the intervention. Both parties filed fewer grievances, and there were fewer staff assaults. The youth who were incarcerated had a decreased incidence of misconduct, restraint, and solitary confinement.

#### 3.3.6. Policy

Policies are used locally, statewide, or federally to “protect the health of a community” [28]; two studies were identified as policy-level interventions that led to an improvement of health and an overall better environment [16,45]. A policy approach to reducing solitary confinement impacted people who were incarcerated and staff [16]. Results showed that solitary confinement was reduced by 74.3% after implementation. Staff reported higher job satisfaction and improved interpersonal interactions and personal safety. Those incarcerated reported that officers used less punitive responses and more problem-solving skills, and reported improved psychological well-being due to less isolation. Both parties perceived increased trust and less antagonism. Implementation of a smoke-free policy resulted in some workers and people who were incarcerated reporting the benefits to their personal health due to less second-hand smoke in the facility [45]. Some respondents commented on an overall better sensory experience. The residents in prison generally acknowledged the health benefits of smoking cessation and improvements in their overall health. 

## 4. Discussion

In this scoping review, identified studies addressed resources that impacted both the correctional workplace and people who are incarcerated, in order to improve safety and health. Findings show that interventions offered at the individual, interpersonal, organizational, or policy level can have a positive health impact on both workers and those incarcerated within the correctional workplace. Research that examines health outcomes for both people who are incarcerated and workers is sparsely recorded. Out of 2545 articles, only 16 articles had measurable outcomes for both.

### 4.1. Individual

Forty-four percent of the studies focused on mental health initiatives for people who were incarcerated. When improving the mental health of those incarcerated, there was intent to decrease antisocial behavior and therefore decrease the workload for workers. Only one article examined the stress levels of both people who were incarcerated and workers, specifically, after a mindfulness intervention [6]. Correctional workers self-report high levels of symptoms of mental disorders, burnout, physical health problems, compassion fatigue, and emotional exhaustion compared to the general population [2,10]. Research shows that work stress is associated with more symptoms of depression in correctional officers and greater depression symptoms are linked to officers mistreating those incarcerated [3]. The impact of these negative health outcomes can affect the greater environment of the workplace for the workers and people who are incarcerated [2]. Research has been completed on improving correctional workplace mental health; however, studies have not examined the influence of workers’ improved mental health on those incarcerated. Further research should examine the interconnectedness of correctional worker mental health and its impacts on people who are incarcerated.

Decreasing punitive responses was a trend in many articles, for example, decreasing forced moves, infractions, program expulsions, recalls, threats, segregation, restraints, and rearrests [16,42,44,49,52,54,55] and instead supplementing with better communication, motivation-based supervision, problem-solving, treatment involvement, trauma-informed approaches, and CBT techniques [16,42,43,44,49,52,54]. When there were fewer punitive responses, those incarcerated felt the worker–person incarcerated relationships were stronger, had more motivation to succeed, and demonstrated improved trust and feelings of safety. Decreasing punitive responses was the main goal of Cloud et al. (2021) [16], with a major reduction in solitary confinement leading to workers having higher job satisfaction, better interpersonal relations, and to overall improved safety levels for staff and people who were incarcerated. Each of the individual-level worker-focused interventions resulted in lowered arrest rates or lowered individual risk of arrest for people who were incarcerated, indicating as well the positive impacts of worker-based interventions for the benefit of people who were incarcerated [43,44,55].

### 4.2. Interpersonal

Across studies, the environments of facilities were reported to improve after the introduction of different kinds of resources. Some of the reasoning for this improvement looked to more positive behaviors, increased feelings of safety, less conflict, and improved interpersonal relations. The relationships between officers and incarcerated persons have been noticed as an essential factor in their achieving success in community outcomes [57]. Research shows that when people who are incarcerated have a supportive and competent relationship with staff and they feel safe, they are then able to achieve better outcomes upon release [58]. Six articles showed an improvement in the relationship between staff and people who were incarcerated; one study, in particular, showed that within a therapeutic community, the officers had more conversations with residents and there was both less misconduct and improved behavior [49]. Offering resources that improve the relationships between staff and those incarcerated has the potential to improve work satisfaction and feelings of safety, and support reentry outcomes.

### 4.3. Organizational and Policy

Organizations, in the manner of correctional facilities, can have positive or negative effects on their members, depending on their organizational characteristics: management, incentives, rules, regulations, or benefits [28]. They have the chance to create norms and values that can improve health, safety, and their environments [28]. Changing the environment of a workplace can take more time than do changes in individual people, but in the end can be more sustainable [59]. Findings from this review show that the majority of the interventions targeted the individual level, which is typically least effective in workplace-health promotion efforts [26]. There is a need for more integrated, multi-level interventions to promote health and safety for both people who are incarcerated and workers. A barrier to creating organizational change and maintaining programs within corrections settings could be attributed to prioritization of security and safety [60]. However, this review has discovered multiple findings that show interventions at all levels decreased violence and aggression [16,46,48,49,51,53,54], and therefore improved safety [16,54].

Eighty-one percent of studies were completed in a community supervision setting or a prison. There was only one study that was completed in a short-term correctional center (jail), which indicates the gap in attention to jail health-promotion research [11]. Understanding what outcomes can be achieved during time in jail may shorten a person’s trajectory into long-term prison sentences. Jails are unique in their position to address acute safety and health situations, as individuals are newly arrested, often facing heightened crisis(es), and held for shorter periods of time as compared to prison settings [61]. Jail work is also distinct from other public safety settings due to its transient nature, lack of personal space, and wide range of different justice-involved backgrounds [62]. Each facility is unique and therefore cannot use a universal approach, but instead needs a tailored approach, specifically, interventions designed using a participatory approach [63]. Due to high rates of recidivism in the United States, there is a trend in criminal justice systems towards implementation of more programming before and after incarceration [53].

In terms of public-health challenges and infectious diseases, there was a focus on improving health-access for people who are incarcerated in conjunction with increasing the knowledge of staff [47,50]. This dual focus allowed workers to better treat and care for residents, because they had more knowledge and skills, while also improving the health of those incarcerated. 

### 4.4. Purposeful Work and Meaningful Activities

Correctional work aims to “provide rehabilitation to those seeking reentry into society” [46]. The findings of this review show the promise of rehabilitative efforts when there are available resources for people who are incarcerated, all workers, or the facility as a whole. When individuals are in stressful living environments, such as correctional facilities, it is important that they strive for a purpose [64], and it is also important for the workers to feel that their work is purposeful. When workers understand the broad purpose of the organization, they are more likely to thrive [65]. This review showed that after implementing aspects of the Norwegian correctional service (i.e, fostering positive relationships, creating community conditions, and less restrictive environments) staff felt they had more purpose [16]. People who were incarcerated felt they had more purpose after participating in a dog training program and a reentry program [48,53]. Utilizing purpose and what people find meaningful is a way to promote health within the correctional world.

The TWH approach is used to meet the mandate by the Occupational Safety and Health Act of 1970 to assure everyone has “safe and healthful working conditions” [59,66]. One platform the TWH approach expands upon is the importance of meaningful work and how that can relate to individual well-being [59]. Research performed by Bailey and Madden (2016) offers insights into what makes work meaningful to people: when their work matters to other people and not just themselves, when they meet challenges and are able to overcome them, and when work is understood in the context of their personal life [65].

It is important to promote engagement in everyday tasks that provide meaning, also known as occupations. Engagement in healthy occupations can enhance health [67]. Therefore, if engagement in work is meaningful and safe for correctional workers, it can be a therapeutic approach to promote health for themselves, and this review shows that it can improve health and safety for people who are incarcerated as well. Similarly, programs that give those incarcerated feelings of purpose can improve the environment of the facility. Research by Worley et al. (2022) [3] found that increased satisfaction gained from work was associated with fewer depressive symptoms in correctional officers and less mistreatment of people who were incarcerated, a finding which is linked to higher levels of depressive symptomatology. Enhancing meaning for workers has the possibility of impacting those incarcerated, and the same can happen for the workers—a study by Bosma et al. (2020) [68] found that the availability of meaningful activities for people who were incarcerated was related to lower self-reported misconduct within a prison. By cultivating meaning and purpose through an occupations-based approach, there is an ability to improve the overall environment and combat occupational deprivation. When people are secluded from opportunities to engage in meaningful occupations there can be long term effects of decreased health and well-being [69,70]. Research shows that improving the environment and implementing occupation-based programs during incarceration can reduce barriers people face during reentry by improving quality of life [71]. Furthermore, use of semi-structured assessment tools for goal-setting can engage people who are incarcerated in goals prioritization and with interventions designed to assist in achieving their goals [72,73].

### 4.5. Promising Studies

While there were only 16 articles addressing integrated approaches to health in correctional settings in this scoping review, nine articles nearly met the criteria and had trends focused on different mental health interventions for people who were incarcerated and workers. A trend with these articles was towards less-punitive responses, and some reported more mental health referrals and de-escalation strategies instead [74,75,76]. Anti-social behaviors such as aggression, self-harm, and misconduct decreased after an intervention was introduced [77,78,79,80,81]. Many promising studies had outcomes of decreased violations or time spent in punishment, which could directly relate to decreasing the workload burden and stress for officers as well, but they did not measure that aspect in their studies. This information could be critical for the justification of making more programming and resources available for both people who are incarcerated and staff. Safer work environments could contribute to worker retention and could be appealing to future employees.

### 4.6. Implications

With most interventions occurring at the individual level, which NIOSH reports as least effective, there is a need for also integrating correctional setting interventions at the organizational and policy level [26]. The goals for these interventions should focus on eliminating hazards and replacing them with health-enhancing resources [26]. While implementing these resources, there should be an examination of the impact on people who are incarcerated, workers, and the environment as a whole using an integrated health and safety approach. Outcomes for both parties should focus on short-term and long-term goals, with the understanding that commitment to an integrated approach can require a long time to understand the total success [26].

The literature supports the need for mental health initiatives, for both people who are incarcerated and workers, to improve the overall environment. There needs to be more focus on how improving workers’ mental health impacts the people who are incarcerated whom they serve. This review also shows that there is low attention to jails, with only one intervention in this setting. This calls for more research on the overall environment in jails, as well as introducing resources in these settings to identify the impact on the health and safety of people and workers in jail.

### 4.7. Limitations and Strengths

This scoping review can be used as a foundation for future research that examines the impact of health and safety resources on correctional facilities; however, some limitations should be considered when interpreting the results. Human error and subjectivity may have influenced data collection, data extraction, and synthesis. Due to the aims of this study focusing on outcomes for both those incarcerated and workers, there is a possibility that there were studies missed if the outcomes were not explicitly stated in the abstract and they were therefore excluded. No critical appraisal of the quality of the studies was conducted, and all study designs were included. Future research should aim to have more definitive outcomes and consider the strength of the evidence. Addressing the multi-level nature of interventions and considering types of preventive strategies are strengths of this study. The authors categorized each article as to subjective best fit, and described the impact of each to inform strategic considerations for systems. This review supports interventions at the individual, interpersonal, organizational, or policy level which address both those incarcerated and workers, in order to improve health and safety for everyone involved.

## 5. Conclusions

This scoping review has outlined the literature on resources that impact people who are incarcerated and correctional worker health. Resources on any level of intervention can be supportive in reducing conflict, improving relations and the environment, and achieving other collateral effects which can improve health and safety for the workers and the people who are incarcerated. At the individual level, findings show that an intervention for those incarcerated has positive implications for workers, and conversely, an intervention for workers has positive implications for people who are incarcerated. The environment of the correctional workplace is interconnected with the residents they serve. Researchers should take steps to understand the impacts of a resource on both populations to understand the full scope and implications of their resource. Leaders within correctional systems should prioritize organizational and policy-level interventions to promote lasting health and safety changes for people who are incarcerated and workers. Since the health and safety needs across correctional systems are complex, the use of an integrated health approach is needed, specifically, one focusing on levels of influence, prevention targets, and the greater long-term impact of the system. This review has shown the paucity of research examining resource implications for both those incarcerated and the workers. The current focus of research has been on improving the health of workers and people who are incarcerated using isolated initiatives. The limited amount of research could be credited to the siloed nature of interventions separated between workforce and incarcerated populations, lacking integrated health approaches. Further research is suggested to understand the full breadth of resources which are the most successful for outcomes for both parties and for the improvement of the correctional system. Increased access to health and safety resources has potential for improvements among these underserved correctional populations.

## Figures and Tables

**Figure 1 ijerph-20-06104-f001:**
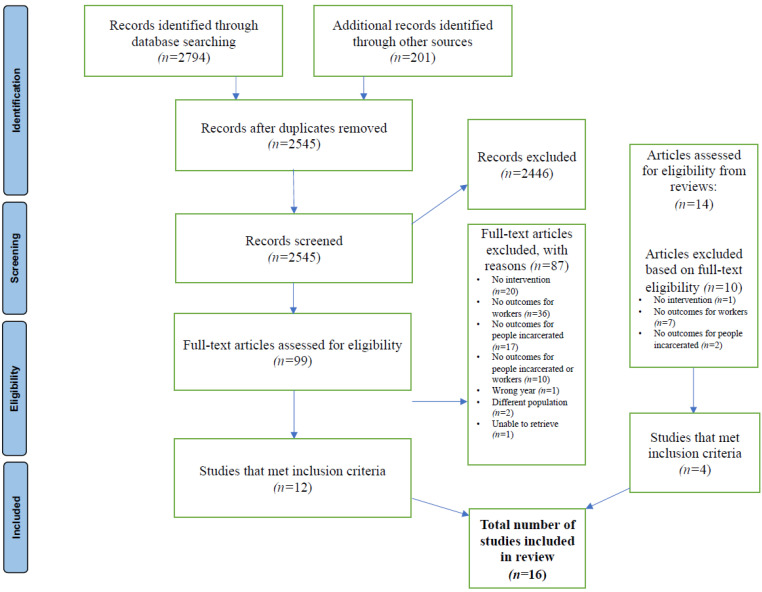
PRISMA Flow Diagram.

**Table 1 ijerph-20-06104-t001:** Scoping Review Inclusion and Exclusion Criteria.

Inclusion	Exclusion
Intervention (program, policy, practice, guideline) targeting workers of correctional facilities and addresses impact of people who are incarcerated OR Intervention (program, policy, practice, guideline) targeting people who are incarcerated and addresses impact of correctional staff	Only addressed correctional staff
Outcomes related to health and safety	Only addressed people who are incarcerated
Full text can be retrieved in English	Not between the years of 2013–2023
Between the years of 2013–2023	Full text can not be retrieved in English
All geographical areas—with the understanding that justice systems around the world are different	Focuses on descriptions of interventions rather than focusing on outcomes
Peer-reviewed (gray literature), including any research study design, thesis/dissertations, discussion papers, or white papers	
Accessible for retrieval	

**Table 2 ijerph-20-06104-t002:** Summary of Studies Addressing Health Promotion Among Correctional Workers and People Who Are Incarcerated.

Author and Year	Context and Population	Aim/Purpose	Outcomes for Workers	Outcomes for Incarcerated People	Significance
Individual—Workers					
Bruce et al., 2017 [55]	Probation. Intervention group: 13 staff, comparison group: 10 staff.	Evaluate the effectiveness of psychologically-informed practice intervention that uses training and consultation to help probation officers work with people who have personality disorders.	Probation staff reported higher levels of knowledge, better understanding of personality disorders, and increased feelings of personal accomplishment compared to the probation staff who did not complete the training.	Probationers had significantly lower rates of recalls and warnings among those who were in the facilities with the intervention compared to the non-intervention facility.	PD-KASQ (self-reporting tool for workers assessing their competency for engaging with people who have personality disorders): Post-intervention (*p* =< 0.01), 6-month follow-up (*p* =< 0.001), 12-month follow-up (*p* =< 0.01). Warning rates and recalls decreased (*p* < 0.01).
Bonta et al., 2021 [43]	Community Corrections. 201 probation officers, 730 clients	Understand if community supervision training on Risk–Need–Responsivity (STICS) is associated with a change in officer behavior, specifically: discussing criminogenic needs, applying behavioral techniques, and understanding how the change impacts clients’ recidivism.	STIC officers discussed attitudes, rather than probation conditions or non-criminogenic needs, with their clients for a significantly higher proportion of each session after training. Collaborative skills, session structuring skills, and cognitive/behavioral techniques significantly improved after training.	General recidivism rates with a 2-year follow-up time were significantly lower for STICs clients, versus a random sample of probationers supervised prior to STICS.	Discussing non-criminogenic needs less: Cohen’s *d* [95% CI]—0.59, [−0.79, −0.39]. Discussing attitudes: Cohen’s *d* [95% CI]—1.01 [0.60,1.42]. Cognitive-Behavioral skills: *Cohen’s d* [95% CI] 1.25 [1.04, 1.47]. Total effective correctional skill Cohen’s *d* [95% CI] 1.26 [1.05,1.48]. Recidivism Rates: 95% CI—0.48, 0.49.
Latessa et al., 2013 [44]	Community Corrections; 21 trained officers and 20 untrained officers; 272 probationers/parolees	Test the effectiveness of teaching community supervision officers Effective Practices in Community Supervision (EPICS) in reducing recidivism for people on supervision.	Community corrections officers who received the training were more likely to focus on criminogenic needs and recognize the link between thoughts and behaviors. Officers who used more of the training techniques had a better perceived relationship with their clients and less re-arrests.	Moderate-risk clients perceived the relationship with their officer to be more caring, fair, and trusting. The high-risk clients found the relationship to be tougher. Clients who perceived a more trusting relationship with their supervisor were significantly less likely to be arrested for a new crime.	Effect of EPICs training on officers using core correctional practices: *p* < 0.001. Clients’ perception of trust and likeliness to be arrested for a new crime: *p* < 0.05.
Individual—People who are incarcerated					
Doyon-Martin and Gonzalez, 2022 [53]	Prison; 97 participants in prison	Understand the benefits of a prison-based dog-program for people in prison to train dogs in four different Midwestern prisons.	Fully 25% of prison staff respondents thought that the program contributed to less violence and improved interactions in the facility; 34% of prison staff respondents believed that this program led to an overall positive attitude within the houses; 10% of prison staff respondents believed it bridged a gap between correctional workers and residents.	A total of 29% of participants in prison reported a rehabilitative impact, 21% reported improved self-esteem, improved parenting skills, and more hope for the future; 32.9% used the exact phrase they felt as if they had a “sense of purpose”; 38% responded that they learned patience from the program and that soft skills will help them with employment in the future; 100% were able to identify at least one personal benefit from the program.	Content analysis of secondary data shows the promising effectiveness of animal socialization programs in prisons.
Miller, 2014 [48]	Correctional Center; 45 people in the correctional center participated	Highlight potential collateral benefits of reentry programming focusing on challenges people who are incarcerated face post-release in a rural reentry initiative.	Reduction in detainee and facility altercations: use of force, fights, staff assaults, and disturbances. Some officers reported that jail culture and operations had improved since the treatment program started.	Reentry participants reported positive thoughts on the program, specifically enjoyed having structure, and said it gave them purpose. Participants reported solidarity between group members.	Quantitative data showed a decrease in altercations, significance not reported.
Jones, 2018 [46]	Prison; four animal-based organizations	Examine animal-based organizations that partner with correctional facilities in implementing prison-based animal programs and understand the impact on people who are incarcerated.	Results indicate perceived improved relations between staff and people in prison—for example, less aggressive communications, and some reported it “bridged a gap” between the two parties.	Perceived impacts on inmates included increased self-esteem, responsibility, self-control, and a positive calm environment. Other outcomes included improved vocational and life skills due to their animal training experience through this program.	Qualitative study, therefore, there was no significant quantitative data to report.
Camp et al., 2018 [51]	Prison; 35 people in prison	Examine the effectiveness of The Enhanced Support Service pilot, which offered individualized psychosocial interventions in reducing violence and disruptions.	From a prison staff perspective, the intervention led to improved coping, understanding and management of behavior, and more pro-social behaviors.	People in prison who used ESS showed decreased levels of aggression and noncompliance, and an increase in positive behaviors post-intervention.	Verbal aggression decreased 54%, *p* < 0.00, physical aggression decreased by −70%, *p* < 0.00, adjudications decreased by −49%, *p* < 0.01, privilege removal decreased by −66%, *p* < 0.00.
Individual—People who are incarcerated and Workers					
Davies et al., 2021 [6]	Prison and Probation. Prison: 21 prison participants, 15 staff; Probation: 28 intervention individuals and 27 control individuals	Investigate the outcomes on staff and prisoners of a brief mindfulness training intervention in a prison and to individuals serving a probation community sentence.	Prison staff’s mindfulness, knowledge, and skills were significantly enhanced and they reported less stress after the intervention.	Participants then in prisons’ mindfulness, knowledge, and skills were significantly enhanced, and they reported less stress after the intervention. Individuals on probation who completed the intervention group showed improvements in mindfulness skills, but the data was not statistically significant.	Combined prisoner and staff total scores on two mindfulness measures between baseline and post-intervention (*p =* 0.006; *p =* 0.024). Combined prisoner and staff perceived stress showed that stress significantly reduced over time (*p* < 0.001).
Neuhaus et al., 2018 [47]	Prison; 173 people who are incarcerated, 16 staff	Explore the clinical effectiveness of Hepatitis C telemonitoring using videoconferences between hepatology specialists and correctional facility clinicians.	Prison clinicians reported improved knowledge and confidence in treating HCV after training. Prison clinicians reported increased workload because they were treating more patients. Prison clinicians reported increased access to care for their patients after implementation of telemonitoring.	A total of 108 incarcerated people commenced treatment for Hepatitis C, with 75% completing the full treatment.	Qualitative study, therefore, no quantitative data to report.
Interpersonal					
Manchak et al., 2014 [42]	Probation; 176 probationers on traditional supervision, 183 probationers on specialty probation services.	To explore the overall effectiveness of specialty probation (program for probationers with mental illness) and understand if specialty officers have better boundary spanning, problem-solving skills, and relationships, and less sanction threats and greater access to mental health services than traditional probational officers.	Specialty officers showed higher boundary spanning and positive compliance, and lower negative compliance strategy than traditional officers. Specialty supervision was more effective in accessing mental health and dual-diagnosis services, and in reducing violation reports.	Probationers with the specialty probation were significantly more likely to receive mental health treatment than were those with traditional probation. Probationers on specialty probation were two times less likely than those on traditional probation to have a formal violation against them. Probationers reported having better relationships and positive compliance strategies than traditional officers.	Probationers receiving mental health treatment: [*p* < 0.001] and integrated dual-diagnosis treatment: [*p* < 0.01]. Specialty probationers met more with their officers [*p* < 0.001] and spent more time with their officers in meetings [*p* < 0.001]. Specialty officers had significantly higher boundary spanning [*p* < 0.001], dual role-relationship [*p* < 0.001], and positive compliance [*p* < 0.01], with lower rates of negative compliance [*p* < 0.01].
Bardin et al., 2022 [52]	Probation; three Probation officers, 113 clients in the first 3 years and 19 more clients during the 6-month extension.	Evaluate the Female Offender Can Ultimately Succeed (FOCUS) program, which focused on meeting women probationers’ needs, improving probation outcomes, and understanding how gender-matching clients and probation officers impacts their success in community transition.	Probation officers indicated that the training helped them acknowledge their clients’ successes. They reported that a decreased caseload allowed them to develop a relationship and restore trust in the system.	Probationers were restored to good probation standing and had their probations terminated at a lower rate than in traditional supervision. Probationers reported that they had good relationships with their officers and that their officers helped them with mental health care, food, employment, and their relationships with their children. They reported that since they could be honest and open with their officers, they could succeed. A total of 50.4% of clients received family welfare services, and 69% received health insurance. Less punitive responses led to more motivation to succeed.	Pilot study, therefore, no significant data to report.
Weinrath et al., 2021 [49]	Prison; therapeutic community 1 = 84 people; therapeutic community 2 = 72 people.	Examine differences in prisoner behavior and the use of alternative resolutions by comparing the misconduct over a 2-year period in therapeutic communities to that of other units.	Therapeutic communities had lower misconduct rates and fewer forced moves than did other units. Officers were more likely to use alternative resolutions in the TC units and reported more interactions that supported stronger relationships.	Therapeutic communities had lower misconduct rates and forced moves than other units. Participants in prison reported that the officers talked and listened more than in the main building, which impacted them favorably, and they reported they are more likely to talk to staff in the TC.	The therapeutic community 2 unit had the lowest misconduct rate (31 charges), and was significantly different from all other units [*p* < 0.05]. Both therapeutic communities had the lowest number of forced moves.
Organizational					
Lichtenstein and Barber, 2016 [50]	Probation/Parole; 8 probation officers and 335 probationers/parolees	Identify if on-site HIV (human immunodeficiency virus) services and training could be implemented successfully for probationers and parolees.	Fully 100% reported having more knowledge about HIV after the education, 88% reported being less fearful about becoming HIV-infected, 63% reported being less concerned about supervising HIV-infected people who were incarcerated, 75% reported that all officers should be educated about HIV, and 100% reported that this program should be permanent.	All 86/86 of the newly incarcerated people received HIV education, 32/86 volunteered for pre-test counseling, and 27/86 volunteered to be tested and receive their results. A total of 29/249 of the currently incarcerated people volunteered for pretest counseling, and 8/249 volunteered to be tested and receive their results.	Pilot study, therefore, no significant data to report.
Elwyn et al., 2015 [54]	Female Juvenile Justice Facility. 2008—26 residents. 2012—30 residents.	Examine the implementation of a trauma informed-intervention (Sanctuary Model) and its impacts on physical and psychological safety for staff and youth at a female juvenile justice facility.	Juvenile staff’s perception of safety improved after the implementation, and the number of grievances and complaints decreased to 0 per 100 staff-days.	Youth who were incarcerated had a decrease in misconduct, solitary confinement, and restraints and reported feeling safer at the facility by a margin of 33%. Youth who were incarcerated filed less grievances towards staff and there were less staff assaults after implementation.	From 2008-2012, youth misconduct decreased from 6.6 to 1.0 (*p* < 0.001), the rate of physical restraints decreased from 7.6 to 1.1 (*p* < 0.001), staff members at the program filed grievances, and complaints decreased from 0.4 to 0 (*p* < 0.05). Youths reporting fearing for their safety decreased from 44% to 11% (*p* < 0.001). Staff perception of safety also improved from 17%, but this was not statistically significant.
Policy					
Brown et al., 2022 [45]	Prison. 99 prison staff and 23 people in prison	Understand the perspectives of prison staff and residents in prison after a comprehensive smoke-free prison policy was implemented.	Prison staff reported the policy was popular due to decreasing secondhand smoke levels and that it was a significant gain for their health. Some prison staff reported experiencing less asthma, sore throat, and eye irritation and commented on an overall better sensory experience after policy implementation.	Some people in prison reported perceived benefits for personal reasons, and non-smokers felt protected from secondhand smoke after policy implementation. Some people in prison commented on an overall better sensory experience and generally acknowledged the health benefits of stopping smoking and improvements in their overall health after policy implementation.	Qualitative study, therefore, no significant data to report.
Cloud et al., 2021 [16]	Prison. Sample of 14 correctional leaders, clinicians, case managers, and line staff; 5 focus groups, with a total of 32 people who are incarcerated.	Understand how changes in solitary confinement are impacting staff and people who are incarcerated in the North Dakota Department of Corrections and Rehabilitation.	Solitary confinement was reduced by 74.28% after policies and practices were implemented. Placements in solitary confinement were shorter (89 days -> 34 days). Placements for people with serious mental illness decreased from 11.39 per month to 1.56—a decrease of 630%. Correctional staff reported improved interpersonal interactions and overall personal safety, as well as higher levels of job satisfaction because their work environment was a more positive place with less conflict and hostility.	Incarcerated people reported that correctional staff used more problem-solving skills and less-punitive responses, and felt there were less assaults, self harm, and violence in the prison after policy/practice implementation. Incarcerated people perceived the reforms as responsible for increased trust, less antagonism between the groups, and improved psychological health and wellbeing as a result of less isolation.	Monthly rates of solitary confinement and overall violent infractions decreased across both prisons: (*p* < 0.05). When the use of solitary confinement decreased, there was a small associated decrease in the monthly rate of violent infractions.

## Data Availability

The datasets used and/or analyzed during the current study are available from the corresponding author upon reasonable request.

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
