# Peer review of "Integrated Safety and Health Promotion among Correctional Workers and People Incarcerated: A Scoping Review"

_ijerph, 2023, doi:10.3390/ijerph20126104_

Round 1

Reviewer 1 Report

This work is interesting, I have found in it a pleasant reading. There is scientific motivation and originality in this manuscript to prove its publication from the method and materials to the research objectives. However, it has to be modified before being considered for publication.

·  The method of “a holistic approach” or “an integrated approach” is mentioned several times in the paper, but it is not clear.

·  The author selected 16 core articles for analysis, and I am curious about the reason for the lack of research on such topics.

The author should provide a detailed explanation of the selection principles for the Eligibility Criteria.

Reviewer 2 Report

thanks for providing an insightful review addressing the safety and health promotion among correctional workers and people incarcerated.

I have added my comments below to help improve the quality of the manuscript for publication:

1- Line 28, you started with the United States without any background. Imagine your reader is from another country. So you narrowed down your audience. Please start with a global approach. For example, an increasing number of people in prison pose a significant threat to the mental health and well-being of workers worldwide (you can use a general reference here). Then talks about USA. USA ranked first in ....

you also mentioned that all geographical areas are considered. So your main focus is not the USA. please start with a statement worldwide.

2-Line 29, replace 6.9 people with 6.9 million people

3-Please avoid qualitatively written words within the entire manuscript. For example, in line 30, "major challenges", instead you can say: face challenges including ..... (the term "major" is not useful).

 4-Line 37: term "high rates" means ???? 2percent or 20 percent? what is classified as "high"?

5- line 40: "detrimental": again qualitative term. In academic papers, it would be better to not stress the statements. For example, comparatively higher mental and physical health problems are reported for incarcerated people (references).

6-line 126: "Selection of Source Evidence" is basically a part of your materials and method. Please move all contents to the end of section 2. For example, Figure 1 and Table 2 is clearly talking about how you screened the articles, what is included and excluded, etc.....

7-Results started from "3.3.1: Individual Workers" line 153

8-After doing an extensive literature review, I expected that you can have numbers about the studies focused on a particular interest. For example, Line 297: instead of "Much of the literature" you can use for example xy% of studies focused on .....

9-again in line 301: Correctional workers self-report high levels: what are high levels? if you compare it to incarcerated people, you can use "comparatively higher levels of .....

10-line 311: Correctional workers are vital in managing prisons and are trained to enforce ...........: is this an implication derived from your results or a general statement? if general statement and you want to keep it, you might wanna put it in your introduction.

11-line 324: what is the larger purpose: you meant the safety of staff and incarcerated people, better interpersonal relationship, .....?

12- line 352: most studies??? 

 13- line 354: the study noted? which study? how did you come up with this study? you were talking about the gap in attention to jail health promotion research and suddenly, a study noted the importance of doing research???

14 suggestions for reviews: you only considered 16 studies in your literature review. The total number of articles that should be considered for a mini-review is around 50-60 articles. So we cannot see a trend in your studies. For examples, during recent years, studies more focused on ..... while ............

Reviewer 3 Report

Congratulations on completing the scoping review entitled “Integrated Safety and Health Promotion Among Correctional Workers and People Incarcerated: A Scoping Review”. Your work conforms almost perfectly to the PRISMA guidance for scoping reviews, which is a remarkable achievement.

It is important to note that although assessment of the methodological quality of the selected studies is not a mandatory requirement in scoping reviews according to PRISMA guidelines, it would be beneficial to explicitly mention this limitation in the article. This would help readers understand that the methodological quality of the studies was not considered in the selection process.

Aside from this consideration, their review provides a comprehensive overview of existing resources and their implications for health and safety in the correctional setting. The conclusions are sound and highlight the importance of taking a comprehensive approach to addressing the challenges in this field.

Overall, their work is solid and has a significant impact in highlighting the need for an integrated approach to improving the health and safety of both correctional workers and incarcerated individuals. It is suggested that they consider mentioning the limitation mentioned above and provide additional recommendations for future research and action.
